# Sequenced-Replacement Sampling for Deep Learning

## Abstract

We propose sequenced-replacement sampling (SRS) for training deep neural networks. The basic idea is to assign a fixed sequence index to each sample in the dataset. Once a mini-batch is randomly drawn in each training iteration, we refill the original dataset by successively adding samples according to their sequence index. Thus we carry out replacement sampling but in a batched and sequenced way. In a sense, SRS could be viewed as a way of performing "mini-batch augmentation". It is particularly useful for a task where we have a relatively small images-per-class such as CIFAR-100. Together with a longer period of initial large learning rate, it significantly improves the classification accuracy in CIFAR-100 over the current state-of-the-art results. Our experiments indicate that training deeper networks with SRS is less prone to over-fitting. In the best case, we achieve an error rate as low as 10.10%.

## 1 Introduction

Stochastic gradient descent (SGD) is an approximation scheme where we only keep a mini-batch in order to reduce the computational cost during the training process. Intuitively, it seems that a smaller mini-batch means a larger approximation, and hence the ultimate model learned must be less accurate. Contrary to the intuition, a relatively smaller mini-batch (above some threshold) could actually outperform a larger one in many applications. The reason behind this counter-intuitive phenomenon is that a relatively smaller mini-batch induces more stochasticity and hence exploration during the training process. It is true that a larger mini-batch may lead to faster convergence, but a relatively smaller mini-batch size could lead to a better accuracy. See Keskar et al. (2016) for more details and discussions. The lesson we learn from the discussions above seems to be that more explorations during the training process may promote generalization. Could we introduce another reasonable type of exploration-inducing mechanism to the training process?

In the deep learning literature, neural networks are typically trained over many epochs. In each epoch, each sample in the dataset is used exactly once. As we enter a new epoch, the samples are reshuffled to form new mini-batches. This means that non-replacement sampling is being used in each epoch. Indeed, non-replacement sampling has been shown to lead to faster convergence than replacement sampling under a relatively general setting Recht & Ré (2012). Training-over-epochs is also recommended in Bengio (2012).

On the other hand, for replacement sampling, a randomly chosen mini-batch is continuously being put back to the original dataset before the next draw of mini-batch. It is conceivable that replacement sampling can lead to more accessible configurations of mini-batches and hence more exploration-induction during the training process. If more explorations during the training process really promote generalization, then it seems that replacement sampling may help to drive the stochastic gradient descent to reach a better local minimum.

However, there is one drawback with replacement sampling — as the randomly chosen mini-batches are continuously being put back to the original dataset before the next random draw, there is no guarantee that all the samples in the dataset would have been utilized with sufficient frequency even after a large number of training steps. This may significantly delay the convergence. Worse still, if a significant amount of samples are not sufficiently touched, it is hard to believe that one could reach a relatively optimal local minimum. To tackle this drawback, we propose a scheme which we call

"sequenced-replacement sampling" for training deep neural networks in this paper. The details of sequenced-replacement sampling will be described in the next section.

We find that sequenced-replacement sampling can significantly improve the classification accuracy in the CIFAR-100 dataset over the current state-of-the-art results. This confirms our intuition that more accessible configurations of mini-batches may induce more explorations during the training process and hence promote generalization.

## 2 SEQUENCED-REPLACEMENT SAMPLING

In the usual training-over-epochs approach, the sampling within one epoch is done without replacement. Suppose that we have $N$ samples and the mini-batch size is $B$. Then the total number of accessible configurations of mini-batches in one epoch is

$$\sum_{k=0}^{n_B-1} \binom{N-kB}{B},$$
(1)

where $n_B = \left[\frac{N}{B}\right]$ is the nearest integer to $N/B$, which also represents the total number of available mini-batches within one epoch. Suppose that we train for $n_E$ epochs before convergence, then the total number of accessible configurations of mini-batches during the entire training process is

$$N_{\text{without}} = n_E \sum_{k=0}^{n_B-1} \binom{N-kB}{B}.$$
(2)

This is also the total number of accessible configurations of mini-batches for $n_E\, n_B$ training iterations.

Alternatively, one could also adopt replacement sampling. In this case, the total number of accessible configurations of mini-batches for $n_E\, n_B$ training iterations is

$$N_{\text{with}} = n_E\, n_B \binom{N}{B} = n_E \sum_{k=0}^{n_B-1} \binom{N}{B}.$$
(3)

One may wonder why we use $\binom{N}{B}$ instead of $\binom{N+B-1}{B}$ which is the well-known formula for the conventional replacement sampling. The reason goes as follows. If we were to adopt the conventional replacement sampling, we would have carried out replacement right after each single sample is drawn. In other words, the replacement would have been carried out on a sample-to-sample basis during the entire mini-batch ($B$ samples) sampling process. However, in our case, we would not carry out replacement until all of the $B$ samples have already been drawn. After the replacement, we start drawing all the next $B$ samples before carrying out the next replacement. Thus we carry out replacement sampling in a batched way, and hence $\binom{N}{B}$ is more relevant.

To compare $N_{\text{with}}$ and $N_{\text{without}}$, we can first consider

$$\frac{\binom{N}{B}}{\binom{N-kB}{B}} = \frac{N\,(N-1)\,\cdots\,(N-B+1)}{(N-kB)(N-kB-1)\cdots(N-(k+1)B+1)}.$$
(4)

Both of the numerator and denominator in the expression above have exactly $B$ terms. Except for the case of $k=0$, each factor in the sequence of factors in the numerator is larger than the corresponding factor in the sequence of factors in the denominator. So we have

$$\binom{N}{B} > \binom{N-kB}{B}, \quad \text{for small } kB,$$
(5)

$$\binom{N}{B} \gg \binom{N-kB}{B}, \quad \text{for large } kB.$$
(6)

For a sufficiently large $(n_B - 1)$, it is obvious that $N_{\text{with}} \gg N_{\text{without}}$.

Thus replacement sampling leads to many more accessible configurations of mini-batches for $n_E\, n_B$ training iterations than the training-over-epochs approach. However, as we pointed out earlier, replacement sampling has a risk of not utilizing all the samples in the dataset with sufficient frequency

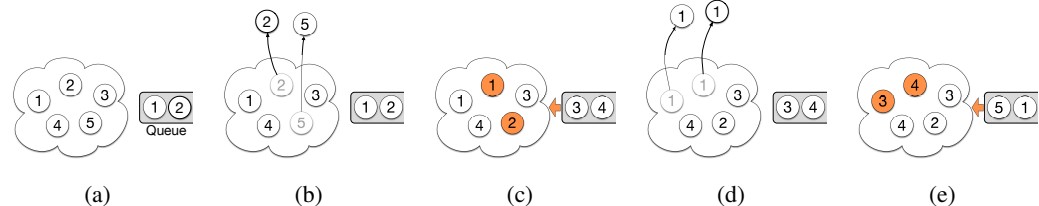

Figure 1: Illustration of sequenced-replacement sampling for mini-batch size $B$=2 and dataset size $N$=5. (a) The queue is filled according to the training example's sequence index. (b) The examples "2" and "5" are sampled from dataset to form a mini-batch. (c) The two examples "1" and "2" from the queue replace the sampled examples "2" and "5" in the dataset. (d) The examples "1" and "1" are sampled to form the next mini-batch. (e) The examples "3" and "4" from the queue replace the sampled examples in the dataset. In the queue, the first indexed example (example "1") comes next to the last indexed example (example "5") iteratively. A queue is employed in this illustration for simplicity, but an array with the indexed dataset and a variable that keeps track of the next index can sufficiently replace it.

even after a large number of training iterations. This may significantly delay the convergence or even lead to a relatively less optimal local minimum.

In this paper, we would like to apply replacement sampling, but we need to tackle the risk of not touching all samples with sufficient frequency. To achieve this goal, we propose "sequenced-replacement sampling", which is illustrated in Figure 1. The idea goes as follows. Before the training process, we first assign a fixed sequence index to each sample in the dataset. As the training process proceeds, a mini-batch is randomly being drawn in each iteration. Instead of putting back the mini-batch that has just been drawn, we refill the original dataset by successively adding samples according to their sequence index. In this way, the samples in the mini-batch that has just been drawn would not have an equal probability to be drawn in the next iteration. In fact, the samples that have not yet been drawn will receive a higher probability to be drawn in the next iteration.

Of course, there is a possibility that exactly the same samples that have just been drawn will be put back to the original dataset through our sequenced-replacement method. However, the chance for such a coincidence is so low that we can ignore it.

With sequenced-replacement sampling, the total number of samples in the dataset remains to be $N$ and the number of samples to be drawn in each training iteration is still $B$. Therefore, the total number of accessible configurations of mini-batches under sequenced-replacement sampling is the same as that under non-sequenced-replacement sampling (the original replacement sampling).

However, the landscape of the accessible configurations of mini-batches under sequenced-replacement sampling will not be the same as that under non-sequenced-replacement sampling. When we apply sequenced-replacement sampling, some accessible configurations of mini-batches under non-sequenced-replacement sampling are no longer accessible. These configurations are mostly replaced by new accessible configurations of mini-batches that are not present under non-sequenced-replacement sampling.

Therefore, we conclude that with sequenced-replacement sampling, the risk of not touching all samples with sufficient frequency is tackled. As discussed above, sequenced-replacement sampling can lead to many more accessible configurations of mini-batches and hence more exploration-induction during the training process. Our conjecture is that more explorations during the training process promote generalization, and our experiments confirm this.

To significantly improve the classification accuracy in the CIFAR-100 dataset over the current state-of-the-art results, we actually need an important trick to facilitate the success of the sequenced-replacement sampling. As we are getting many more accessible configurations of mini-batches and hence more exploration-induction, we need the learning rate $\lambda$ to maintain the initial large value for a longer period of time. The reason is that mere existence of many more accessible configurations of mini-batches is not sufficient, we may also require a sufficiently large learning step to actually visit various good local minima and find the optimal one. It is the combination of more accessible con-

figurations of mini-batches and a sufficiently large learning rate that leads to effective explorations. In fact, this can also be understood mathematically. First of all, with the gradient descent method, each weight (model parameter) at time $t$, $\mathbf{w}_i^{(t)}$, is updated with the following formula:

$$\mathbf{w}_i^{(t+1)} = \mathbf{w}_i^{(t)} - \lambda \left. \frac{\partial \mathcal{L}}{\partial \mathbf{w}_i} \right|_{\mathbf{w}_i = \mathbf{w}_i^{(t)}} \quad , \tag{7}$$

where $\mathcal{L}$ is the loss function. The accessible configurations of mini-batches affect the factor $\left. \frac{\partial \mathcal{L}}{\partial \mathbf{w}_i} \right|_{\mathbf{w}_i = \mathbf{w}_i^{(t)}}$ which dictates the vectorial direction of the gradient descent. More accessible configurations of mini-batches induce more stochasticity through $\left. \frac{\partial \mathcal{L}}{\partial \mathbf{w}_i} \right|_{\mathbf{w}_i = \mathbf{w}_i^{(t)}}$. But it is through a sufficiently large learning rate that the weight updates may be given enough momentum to reach various good local minima.

## 3 EXPERIMENTS

### 3.1 EXPERIMENT DESIGN

|  | CIFAR-10 | CIFAR-100 |
| --- | --- | --- |
| Classes | 10 | 100 |
| Resolution | 32×32 | 32×32 |
| Channels | 3 | 3 |
| Training images per class | 5,000 | 500 |
| Test images per class | 1,000 | 100 |

Table 1: Dataset statistics.

In order to validate the effectiveness of sequenced-replacement sampling, we apply it to two of the state-of-the-art methods. Then, we experiment them on public datasets and compare them with state-of-the-art methods where the common non-replacement sampling is applied. We report error rate of classification on test data and show plots of training loss and test accuracy for different epochs.

We choose public datasets CIFAR-10 and CIFAR-100 Krizhevsky & Hinton (2009) for experiments. The statistics of the datasets are described in Table 1. As shown in the table, there are much fewer training images per class in CIFAR-100 than in CIFAR-10. For pre-processing, we normalize the data with each channel's mean and stardard deviation. Then, we apply horizontal random flipping and random cropping (from image padded by 4 pixels on each side) to training images in each batch.

We train two of the state-of-the-art architectures, Wide Residual Networks (*Wide ResNet*) Zagoruyko & Komodakis (2016) and Densely Connected Convolutional Networks (*DenseNet*) Huang et al. (2016a), with our proposed sequenced-replacement sampling (**WRN-SRS** and **DN-SRS**).

*Wide ResNet.* This network has been proposed to expedite the training in deep residual networks He et al. (2016a), whose training can be very slow due to the depth of the network. Emphasizing the width of the network, instead of the depth, 16-layer-deep Wide Residual Networks were able to outperform thousand-layer-deep residual networks. Not only the architecture is more computationally efficient than residual networks by decreasing the depth of the network, but also it is regarded as one of the state-of-the-art in terms of its effectiveness.

*DenseNet.* This network has been proposed to solve the vanishing gradient and input problems. In the architecture, each layer is connected to every other layer in a feed-forward fashion, in order to ensure maximum information flow between layers. As a result, its effectiveness is regarded as one of the state-of-the-art while it requires relatively little computation.

As the two architectures have been shown to be very effective in image classification, we embed our sequenced-replacement sampling approach in them. However, our method is general and not limited to Wide ResNet and DenseNet. In addition, as our method does not change the underlying architectures, the number of the parameters remains the same. Also, our method does not require additional hyper-parameters, so no extra effort on training process is needed.

An important clarification is required here. In addition to the original residual unit proposed in He et al. (2016a), the authors explored a few similar configurations with different orderings of batch normalizations and activation functions He et al. (2016b). These include "BN after addition", "ReLU before addition", "ReLU-only pre-activation", and "full pre-activation" (see Figure 4 in He et al. (2016b)). By **"WRN-SRS"**, we actually refer **"WRN"** to the Wide ResNet with the "BN after addition" residual unit, not the original one. The "BN after addition" residual unit is unique in the sense that it provides an "overall" batch normalization after the addition. As we anticipate, SRS leads to much more fluctuations, and hence significantly more covariate shift. In order for SRS to bring constructive effects and cause minimal covariate shift, we need an "overall" batch normalization after the addition. Indeed our experiments have confirmed that SRS correlates much better with the "BN after addition" residual unit than the original one. We reserve the terminology **"Wide ResNet"** for the Wide ResNet with the original residual unit.

As for DenseNet, it basically uses something like the "full pre-activation" residual unit, not the original one either. However, it has three "overall" batch normalization units for the entire network to minimize the covariate shift caused by SRS. Thus, for **"DN-SRS"**, we keep the original DenseNet architecture, except that the non-replacement sampling is replaced by SRS.

For sequenced-replacement sampling, the "number of epochs" is not well defined. In order to have a more intuitive comparison with non-replacement sampling, we define an "effective epoch" for sequenced-replacement sampling:

$$\text{effective epoch for SRS} = \frac{n_I}{n_{\text{one epoch}}} \; , \tag{8}$$

where $n_I$ is the number of iterations completed using SRS and $n_{\text{one epoch}}$ is the number of iterations within one epoch using non-replacement sampling. This would be what we mean by "the epochs for SRS" from now on.

We use the following configuration of the original Wide ResNet. The architecture's depth is set to 28 and the widening factor $k$ is set to 10 unless specified otherwise. The dropout Srivastava et al. (2014) is also adopted with the keep probability of 0.7, and weight decay of 0.0005 is applied to all weights. Stochastic gradient descent with momentum of 0.9 is adopted as the optimizer, and the mini-batch size is set to 64.[1] The initial learning rate is set to 0.1 and decays by a factor of 10 at the epochs (120, 150, 175) unless specified otherwise. The models are trained for 200 epochs in total. For DenseNet, we set its depth to 190 and its growth rate to 40. The same configuration of the mini-batch size, weight decay, and the learning rate schedule is applied to DenseNet. Tensorflow is adopted to implement our method, and all experiments are done with a single GPU, Nvidia Titan X.

## 3.2 EXPERIMENT RESULTS

The test error rates of our method and baselines are shown in Table 2. With 28 layers, WRN-SRS achieves an error rate of 12.34% on CIFAR-100. Clearly, it outperforms the original Wide ResNet, WRN and DenseNet as well as all other baselines on CIFAR-100. Indeed, WRN-SRS has achieved a 22% relative improvement over the state-of-the-art method, Shake-Shake Regularization (15.85%). It is also a 36% relative improvement over the underlying method, WRN. DN-SRS achieves an error rate of 15.63% on CIFAR-100, which is a 1.4% relative improvement over Shake-Shake Regularization and 9.0% over the original DenseNet.

Comparing the classification error rates for CIFAR-100 achieved by the methods "Wide ResNet (19.25%)", "Wide ResNet with SRS (19.05%)", "WRN (19.42%)", and "WRN-SRS (12.34%)", it is clear that Wide ResNet works slightly better than WRN. However, SRS correlates much better with the "BN after addition" residual unit than the original one, and WRN-SRS significantly outperforms "Wide ResNet with SRS".

What is most remarkable is that with 70 layers, our WRN-70-SRS achieves an error rate as low as 10.10% in the CIFAR-100 dataset. But WRN-70 achieves an error rate of 19.72% which is slightly worse than the 19.42% achieved by WRN (28 layers). This implies that solely adding more layers to WRN without SRS may actually overfit the CIFAR-100 dataset. In contrast, training with SRS

---

[1]Notice that the original Wide ResNet is trained with a mini-batch size of 128, but we observed that either a size of 64 or 128 did not lead to significant difference for WRN. Please see the following section for details.

| Method | | Depth(-$k$) | Params. | C-10 | C-100 |
|---|---|---|---|---|---|
| Network in Network | Lin et al. (2013) | | | 8.81 | 35.67 |
| All-CNN | Springenberg et al. (2014) | | | 7.25 | 33.71 |
| Deeply-Supervised Net | Lee et al. (2015) | | | 7.97 | 34.57 |
| FitNet | Romero et al. (2014) | | | 8.39 | 35.04 |
| Highway Networks | Srivastava et al. (2015) | | | 7.72 | 32.39 |
| Exponential Linear Units | Clevert et al. (2015) | | | 6.55 | 24.28 |
| ResNet | He et al. (2016a) | 110 | 1.7M | 6.43 | 25.16 |
| | | 1202 | 10.2M | 7.93 | 27.82 |
| ResNet with Stochastic Depth | Huang et al. (2016b) | 110 | 1.7M | 5.23 | 24.58 |
| | | 1202 | 10.2M | 4.91 | - |
| Shake-Shake Regularization | Gastaldi (2017) | 26 | 26.2M | **2.86** | - |
| | | 29 | 34.4M | - | 15.85 |
| DenseNet | Huang et al. (2016a) | 190-40 | 25.6M | 3.46 | 17.18 |
| DN-SRS | **ours** | 190-40 | 25.6M | - | 15.63 |
| Wide ResNet | Zagoruyko & Komodakis (2016) | 40-4 | 8.9M | 4.53 | 21.18 |
| | | 16-8 | 11.0M | 4.27 | 20.43 |
| | | 28-10 | 36.5M | 4.00 | 19.25 |
| Wide ResNet with SRS | **ours** | 28-10 | 36.5M | 4.18 | 19.05 |
| WRN | **ours** | 28-10 | 36.5M | 4.37 | 19.42 |
| WRN-SRS | **ours** | 28-10 | 36.5M | 4.06 | **12.34** |
| WRN-70 | **ours** | 70-10 | 106.4M | 4.49 | 19.72 |
| WRN-70-SRS | **ours** | 70-10 | 106.4M | 4.36 | **10.10** |

Table 2: Test error rate (%) of different methods on CIFAR-10 and CIFAR-100. The widening factor and the growth rate are denoted next to depth for WRN-based models and DenseNet-based models, respectively. The error rate of our method is median of five runs.

allows us to add more layers to WRN and improve the accuracy further. Training deeper networks with SRS not only does not cause overfitting, but also improves generalization. Notice that WRN-70-SRS has achieved a 36% relative improvement over the state-of-the-art method, Shake-Shake Regularization (15.85%).

Interestingly, although our method largely outperforms existing methods on CIFAR-100, it does not outperform them on CIFAR-10. As shown in Table 1, CIFAR-10 contains 5,000 training images-per-class as opposed to 500 images-per-class in CIFAR-100. With a relatively sufficient images-per-class in the CIFAR-10 dataset, the local minima achieved by good neural network architectures, such as Shake-Shake Regularization, may already be so optimal that it is difficult for the explorations provided by sequenced-replacement sampling to surpass them. Since sequenced-replacement sampling introduces more stochasticity, there is a possibility that a more fine-tuned learning-rate schedule is needed to properly realize its true constructive effect on the CIFAR-10 dataset.

In fact, SRS is not without any constructive effect on the CIFAR-10 dataset when it is embedded within the wide resnet architecture. For instance, Table 2 indicates that WRN-SRS and WRN-70-SRS work respectively better than WRN and WRN-70. Of course, since SRS does not correlate well with the original residual unit, it is not surprising that "Wide ResNet with SRS" works slightly worse than "Wide ResNet" for CIFAR-10.

One may think that our improvement actually comes from using only a smaller mini-batch size or keeping the high initial learning rate for a longer period than WRN. In order to make a fair comparison between non-replacement sampling and sequenced-replacement sampling, we train WRN, *i.e.*, with non-replacement sampling, from scratch with configurations that match ours. The experiment results are shown in Table 3. Firstly, we experimented if the mini-batch size significantly affects the result of non-replacement sampling. Using the mini-batch size of 64 or 128 with the original learning rate schedule does not make significant difference, which means that the mini-batch size cannot be the factor of our improvement. Then, we trained WRN with non-replacement sampling in configuration that matches both mini-batch size and learning rate schedule of our sequenced-replacement sampling. Again, the performance of WRN does not improve, but it rather degrades. These empirical results substantiate that our sequenced-replacement sampling approach is indeed effective and the improvement does not merely originate from the mini-batch size and learning rate

| Method | $B$ | LR schedule | Decay | Error |
|---|---|---|---|---|
| | 128 | (60, 120, 160) | 0.2 | 18.38 |
| WRN | 64 | (60, 120, 160) | 0.2 | 18.47 |
| | 64 | (120, 150, 175) | 0.1 | 19.42 |
| WRN-SRS | 64 | (120, 150, 175) | 0.1 | 12.34 |

Table 3: Performance of WRN and WRN-SRS on CIFAR-100 with different configurations. $B$ denotes mini-batch size. Learning rates decay at the epochs in "LR schedule with" by the factor of "Decay".

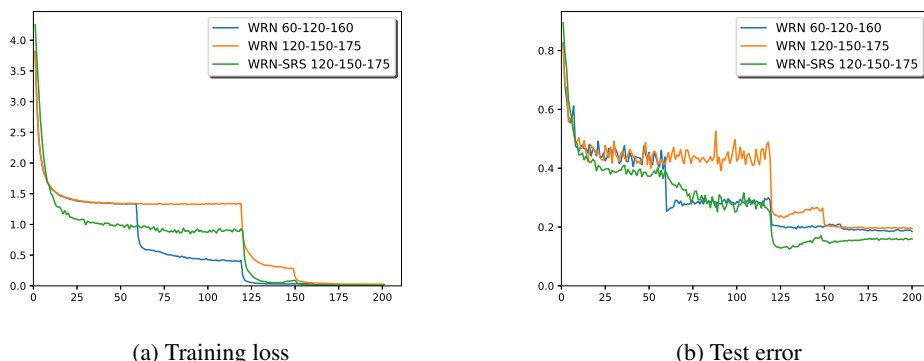

(a) Training loss  (b) Test error

Figure 2: Training loss and test error of WRN and WRN-SRS on CIFAR-100. The three models correspond to the second, third, and fourth model in Table 3.

schedule but from the combination of sequenced-replacement sampling and a longer period of initial large learning rate.

The training loss and test error of our model as well as WRN on CIFAR-100 are depicted in Figure 2. Although the same initial learning rate is applied to both non-replacement sampling (WRN) and sequenced-replacement sampling (WRN-SRS; ours), we observe that our method reaches a lower training loss earlier than WRN (Figure 2a). But as expected, non-replacement sampling converges faster than sequenced-replacement sampling. In fact, the training loss of our method keeps decreasing after 60 epochs, while the training loss of WRN 120-150-175 converges earlier and doesn't seem to decrease until 120th epoch. Interestingly, during this period, although the training loss decreases very slowly with our method, the test error decreases relatively quick so that the error becomes as low as that of WRN 60-120-160, which has already lowered the learning rate at 60th epoch (Figure 2b). On the other hand, the test accuracy of WRN 120-150-175 does not decrease after 60th epoch unlike ours.

In order to closely see the effect of the longer period of initial large learning rate on test error, we show, in Figure 3, the training loss and test error of sequenced-replacement sampling with different learning rate schedules. As shown in Figure 2, although the training loss decreases very slowly after 60th epoch, the error rate decreases relatively quick. Would decaying learning rate earlier than 120th epoch significantly affect the performance? It is shown, in Figure 3b, that the extended period of the initial large learning rate is indeed crucial for our sequenced-replacement sampling. Decaying the learning rate early at 60th epoch leads to the final best error rate of about 23%, which is far higher than 12.34% obtained with the extended period of initial large learning rate.

Meanwhile, we notice a disturbing effect in training after the learning rate decays, where the training loss and test error slightly increase. It has been also noticed in the original Wide ResNet Zagoruyko & Komodakis (2016), and it is known to be caused by weight decay. We look forward to understanding it better in the forthcoming work.

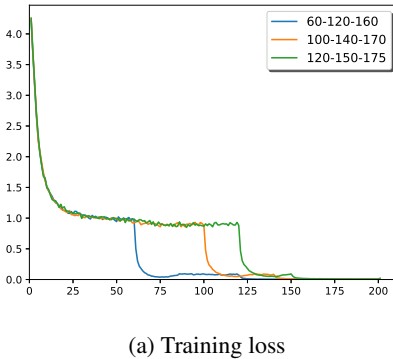

(a) Training loss

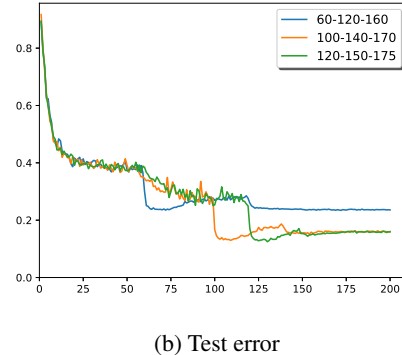

(b) Test error

Figure 3: Training loss and test error of WRN-SRS on CIFAR-100 with different learning rate schedules. The mini-batch size is set to 64. The learning rate decays by a factor of 10 according to each schedule.

## 4 RELATED WORK

*Stochastic Gradient Descent and Generalization.* It has been reported that the noisy gradient from stochastic learning can result in better solutions than batch learning for neural networks, because the noise can let the weights jump across different basins of local minima while weights found by batch learning stays in the same basin as the initial weights LeCun et al. (1998). It has been also observed that using a larger mini-batch tends to converge to sharp minima and thus poorer generalization, whereas using a smaller mini-batch tends to converge to flat minima that lead to better accuracy Keskar et al. (2016). Such observation has led researchers to propose Entropy-SGD Chaudhari et al. (2016), in which local-entropy-based objective function favors solutions in flat regions of the energy landscape. On the other hand, it has also been claimed that a more careful definition of flatness is needed because reparameterization of parameter space can turn a minimum significantly flatter by obtaining equivalent parameter with large Hessian eigenvalues Dinh et al. (2017).

*Replacement and Non-replacement Samplings for Stochastic Gradient Descent.* Researchers have compared the replacement and non-replacement samplings for stochastic gradient descent especially in convex optimization setting, and it has been found that non-replacement sampling often achieves better empirical performance in general. Bottou observed that non-replacement sampling converges much faster than replacement sampling Bottou (2009). Later, it has been theoretically analyzed that non-replacement sampling leads to faster expected convergence rate Recht & Ré (2012); Gürbüzbalaban et al. (2015). Faster convergence of non-replacement sampling has also been observed and proved for non-convex distributed stochastic gradient descent Meng et al. (2017) recently.

## 5 DISCUSSIONS

Undoubtedly, SRS leads to slower convergence than the training-over-epochs (non-replacement sampling) approach. However, since we do not need SGD and mini-batch sampling during the inference, the extra computational cost introduced by SRS (through slower convergence) is only affecting the training process. In other words, SRS promotes generalization, but it does not introduce extra computational cost to the inference. This is a unique feature because many other generalization-promoting schemes (such as batch normalization, layer normalization, ELU, etc) do introduce extra computational cost to the inference even though they may accelerate the convergence during the training process. As a consequence, SRS is especially useful for training a more accurate model to be deployed in embedded devices. In addition, SRS does not introduce any additional parameters or hyper-parameters, and this makes it even more attractive.

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
