# OpenReview forum: "Sequenced-Replacement Sampling for Deep Learning"
_ICLR.cc/2019/Conference_

### Official Review · AnonReviewer2 · 2018-10-31
**Needs More Work**

**Rating:** 4
**Confidence:** 4

**Review:**

In this paper, the authors introduce a sampling strategy that aims to combine the benefits of with- and without-replacement SGD. With-replacement strategies add more randomness to the process, which the authors claim helps convergence, while the without-replacement strategies ensure equal usage of all datapoints. The authors present numerical results showing better convergence and improved final accuracy. While I found the idea appealing, I felt that the paper needs more work before it can be published. I detail some of my primary concerns below:

- The entire motivation of the paper is predicated on the hypothesis that more randomness is better for training. This is not generally true. Past work has shown that specific kinds of random noise aid convergence through exploration/saddle point avoidance/escaping spurious minima while others either make no change, or hurt. Noise from sampling tends to be a structured noise that aids exploration/convergence over batch gradient descent, but it is not immediately clear to me why the choice between with- and without-replacement should imply exploration.

- Maybe it's obvious but I'm not grasping why, mathematically, the number of accessible configurations for SRS is the same as original replacement sampling (4th paragraph on page 3).

- Given that the central motivation of the work was to enable with-replacement strategies while still ensuring equal usage, I recommend that the authors include a histogram of datapoint usage for three strategies (with, without, hybrid). This should help convince the reader that the SRS indeed improves upon the usage statistics of with replacement.

- If one were to create a hybrid sampling strategy, one that is natural is doing 50-50 sampling with and without replacement. In other words, for a batch size of 64, say, 32 are sampled with replacement and 32 without. By changing the ratio, you can also control what end of the sampling spectrum you want to be on. Did you try such a strategy?

- For the numerical experiments, as I see it, there are 3 differences between the SRS setup and the baseline: location of batch normalization, learning rate, and batch size. The authors show (at the bottom of Page 6) that the performance boost does not come from learning rate or mini-batch size, but what about the placement of the BN layer? Seems like that still remains as a confounding factor?

- "SRS leads to much more fluctuations, and hence significantly more covariate shift". How do the authors define covariate shift? Can the authors substantiate this claim theoretically/empirically?

- The authors claim that the method works better when the dataset size is low compared to number of classes. Again, can the authors substantiate this claim theoretically/empirically? Maybe you can try running a sub-sampled version of CIFAR-10/100 with the baselines?

- The writing in the paper needs improving. A few sample phrases that need editing: "smaller mini-batch means a larger approximation", "more accessible configurations of mini-batches", "hence more exploration-induction", "less optimal local minimum"

- Minor comment: why is the queue filled with repeated samples? In Figure 1, why not have the system initialized with 1 2 3 in the pool and 4 5 in the queue? Seems like by repeating, there is an unnecessary bias towards those datapoints.

---

### Official Review · AnonReviewer3 · 2018-11-02
**The approach requires more validation**

**Rating:** 5
**Confidence:** 5

**Review:**

This paper constructs a very simple, new scheme for sampling mini-batches. It aims to (i) achieve the noise properties of sampling with replacement while (ii) reduce the probability of not touching a sample in a given epoch. The result is a biased sampling scheme called “sequenced-replacement sampling (SRS)”. Experimental results show that this scheme performs significantly better than a standard baseline on CIFAR-100 with minor improvements on CIFAR-10.

This is a highly empirical paper that presents a simple and sound method for mini-batch sampling with impressive results on CIFAR-100. It however needs more thorough analysis or experiments that validate the ideas as also experiments on harder, large-scale datasets.

Detailed comments:

1. The authors are motivated by the exploration properties of sampling with replacement which I find quite vague. For instance, https://arxiv.org/abs/1710.11029 , https://arxiv.org/abs/1705.07562 etc. show that sampling mini-batches with replacement has a large variance than sampling without replacement and consequently SGD has better regularization properties. Also, for mini-batches sampled with replacement, the probability of not sampling a given sample across an epoch is very small.

2. I believe the sampling scheme is unnecessarily complicated. Why not draw samples with replacement with a probability p and draw samples without replacement with a probability 1-p? Do the experimental results remain consistent with this more natural sampling scheme which also aligns with the motivations of the authors?

3. To validate the claim that SRS works well when there are fewer examples per class, can you do ablation experiments on CIFAR-10 or ImageNet/restricted subset of it?

---

### Official Review · AnonReviewer1 · 2018-11-04
**More work is needed**

**Rating:** 3
**Confidence:** 4

**Review:**

The paper suggests a new way of sampling mini-batches for training deep neural nets. The idea is to first index the samples then select the batches during training in a sequential way. The proposed method is tested on the CIFAR dataset and some improvement on the classification accuracy is reported.

I find the idea interesting but feel that much more is needed in order to have a better understanding of how the proposed method works, when it works and when it doesn't work. Some theoretical insight, or at least a more systematic experimental study, is needed to justify the proposed method.

---

### Meta-Review · Area_Chair1 · 2018-12-13

**Confidence:** 4
**Recommendation:** Reject

**Metareview:**

This paper proposes a new batching strategy for training deep nets. The idea is to have the properties of sampling with replacement while reducing the chance of not touching an example in a given epoch. Experimental results show that this can improve performance on one of the tasks considered. However the reviewers consistently agree that the experimental validation of this work is much too limited. Furthermore the motivation for the approach should be more clearly established.